# Photocatalytic aerobic oxidation of C(sp³)-H bonds

Lei Zhang[1,3], Run-Han Li[1,3], Xiao-Xin Li[1]✉, Shengyao Wang [2], Jiang Liu [1], Xiao-Xuan Hong[1], Long-Zhang Dong [1], Shun-Li Li[1] & Ya-Qian Lan [1]✉

In modern industries, the aerobic oxidation of C(sp³)-H bonds to achieve the value-added conversion of hydrocarbons requires high temperatures and pressures, which significantly increases energy consumption and capital investment. The development of a light-driven strategy, even under natural sunlight and ambient air, is therefore of great significance. Here we develop a series of hetero-motif molecular junction photocatalysts containing two bifunctional motifs. With these materials, the reduction of O₂ and oxidation of C(sp³)-H bonds can be effectively accomplished, thus realizing efficient aerobic oxidation of C(sp³)-H bonds in e.g., toluene and ethylbenzene. Especially for ethylbenzene oxidation reactions, excellent catalytic capacity (861 mmol g cat⁻¹) is observed. In addition to the direct oxidation of C(sp³)-H bonds, CeBTTD-A can also be applied to other types of aerobic oxidation reactions highlighting their potential for industrial applications.

In industrial oxidation processes with a demand of >1000 tons/annum listed in Ullmann's Encyclopedia of Industrial Chemistry, the direct oxidation of C(sp³)-H bonds using O₂ as an oxidant agent accounts for a significant proportion (49 examples of 109 total examples)[1,2]. The obtained products have a wide range of applications in the pharmaceutical and fine chemical industries[3-8]. However, such oxidation processes are currently performed under high temperatures and pressures in the industry (commonly >100 °C, >5 bar) (Fig. 1a). Such harsh conditions cause not only high demands on the equipment and significant energy losses but also many safety issues (e.g., exothermic reactions, flash points, spontaneous combustion). The development of light-driven catalytic systems at ambient temperature and pressure to realize this process efficiently would be of great importance[9-14]. In particular, the photocatalytic systems under sunlight and ambient air possess the advantages of energy saving, simple reaction conditions, safety, and manageability, which is more in line with the requirements of green and sustainable chemistry[15,16]. In an ideal photocatalytic system, the aerobic oxidation of C(sp³)-H bonds requires photogenerated electrons and holes from the catalyst to complete the activation of O₂ and C(sp³)-H bonds, respectively, and further complete the whole oxidation reaction (Fig. 1b)[17]. It requires that the photocatalyst has active centers for both O₂ and C(sp³)-H bond activation. Moreover, the

two active sites need to be preceded by a close distance that facilitates the separation of the photogenerated charges and the rapid mass transfer of the reaction intermediates[18,19]. Based on such considerations, heterojunctions have been constructed in recent years to help separate photogenerated electrons and holes[20,21]. However, the limited contact area of their structural components and the inhomogeneous distribution of active sites not only limit the further improvement of photocatalyst performance but also make it difficult to investigate clear structure-activity relationships at the molecular level. Therefore, it is important and necessary to design and develop efficient photocatalysts with well-defined structures to realize aerobic oxidation of C(sp³)-H bonds under mild conditions[22,23].

Combining the design idea of heterojunctions with crystalline coordination polymers to synthesize hetero-motif molecular junctions allows for both the construction of bifunctional photocatalysts and also has clear structural information to study the reaction mechanism[24]. In this regard, co-assembling structural units with the abilities of light-driven O₂ and C(sp³)-H bond activation to construct hetero-motif molecular junction photocatalysts would be a viable strategy for realizing photocatalytic C(sp³)-H bond oxidation reactions. In the expected hetero-motif molecular junction photocatalysts, the two active motifs are fixed in the crystalline structure by strong

[1]School of Chemistry, South China Normal University, Guangzhou 510006, P. R. China. [2]College of Science, Huazhong Agricultural University, Wuhan 430070, P. R. China. [3]These authors contributed equally: Lei Zhang, Run-Han Li. ✉e-mail: xxli@m.scnu.edu.cn; yqlan@m.scnu.edu.cn

## Aerobic Oxidation of C(sp³)−H Bonds

**a**. Traditional industrial synthesis method

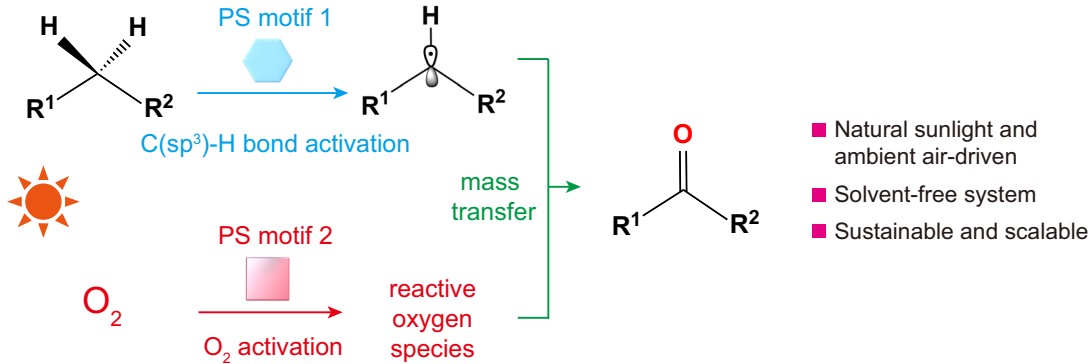

**b**. Desired photocatalytic synthesis method

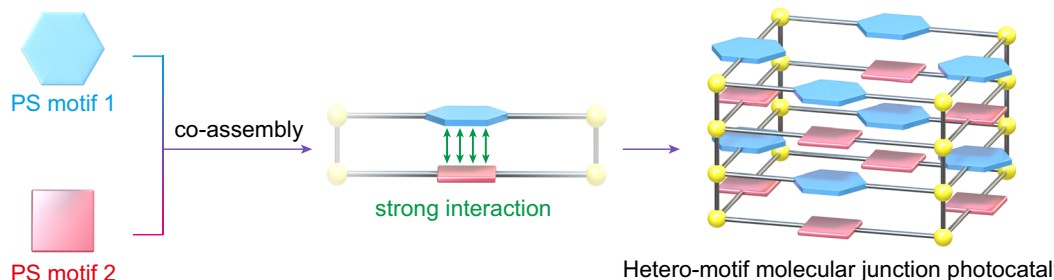

**c**. Design ideas of photocatalyst (this work)

**Fig. 1 | Catalysts for direct oxidation of C(sp³)−H bonds. a** Industrial aerobic oxidation process using transition-metal-catalysts. **b** Aerobic oxidation processes in a desired photocatalytic synthesis method. **c** Design ideas of hetero-motif molecular junction photocatalyst for photocatalytic aerobic oxidation of C(sp³)−H bonds.

interaction (coordinate bond, covalent bond van der Waals forces, etc.). The motifs can be photoexcited simultaneously when light is illuminated. The strong interaction is conducive to the rapid photo-generated charge migration between different components to form separated electron–hole pairs, realizing the efficient charge separation and further completing the activation of $O_2$ and C(sp³)−H bonds. Meanwhile, the relatively short spatial distance between active motifs is beneficial to the mass transfer of reactive intermediates, effectively avoiding the active intermediates quenching and finally completing the oxidation of C(sp³)−H bonds rapidly (Fig. 1c). The modifiability of crystalline materials can contribute to adjusting the types of active motifs, light absorption range, strength of the intermolecular force and spatial distance. The periodic structures make the active sites evenly dispersed in space, which is conducive to further improving the catalytic performance[25,26]. Therefore, selecting suitable components and arranging them to assemble the hetero-motif molecular junction photocatalysts are the keys to completing the aerobic oxidation of C(sp³)−H bond reactions.

Based on the above consideration, a series of hetero-motif molecular junction photocatalysts (CeBTTD-B CeBTTD-N and CeBTTD-A) were assembled with a perylene diimide ligand (H₄BTTD) (Supplementary Fig. 1) and three conjugated ligands (benzoic acid

(BA), 1-naphthalene acid (NA) and 9-anthroic acid (9-AC)). Due to the restriction of the coordination orientation of the Ce₂ secondary building unit (SBU), the BTTD⁴⁻ and the secondary ligands exhibit short spatial distances (3.3–3.8 Å) and strong π-π interaction in the structure of CeBTTD photocatalysts. When the three photocatalysts were used for the aerobic oxidation of C(sp³)−H bond reactions, CeBTTD-A showed high efficiency in the oxidation of toluene to benzoic acid and ethylbenzene to acetophenone with yields in excess of 98%. Structural analysis, experimental characterization, and theoretical calculation results demonstrate that benefiting from the good light absorption and the strong interaction between BTTD⁴⁻ and 9-AC, the two motifs reveal bicomponent excitation and promote the rapid charge transfer and separation to form the electron–hole pairs of BTTD·⁻ and 9-AC·⁺ under light illumination. The motifs in the excited state can quickly complete oxygen reduction to superoxide radical ($O_2·^−$) and C(sp³)−H bond activation. In addition to the aerobic oxidation of C(sp³)−H bond reactions, CeBTTD-A can also be applied to other oxidation reactions (benzyl alcohol oxidation, anisole oxidation, and benzylamine oxidation coupling reaction). The yields of over 20 obtained products are above 90%. It is worth noting that CeBTTD-A can complete a solvent-free large-scale (10 g) ethylbenzene oxidation, and the catalytic activity (861 mmol $g_{cat}^{-1}$ with 91.4% yield), which is

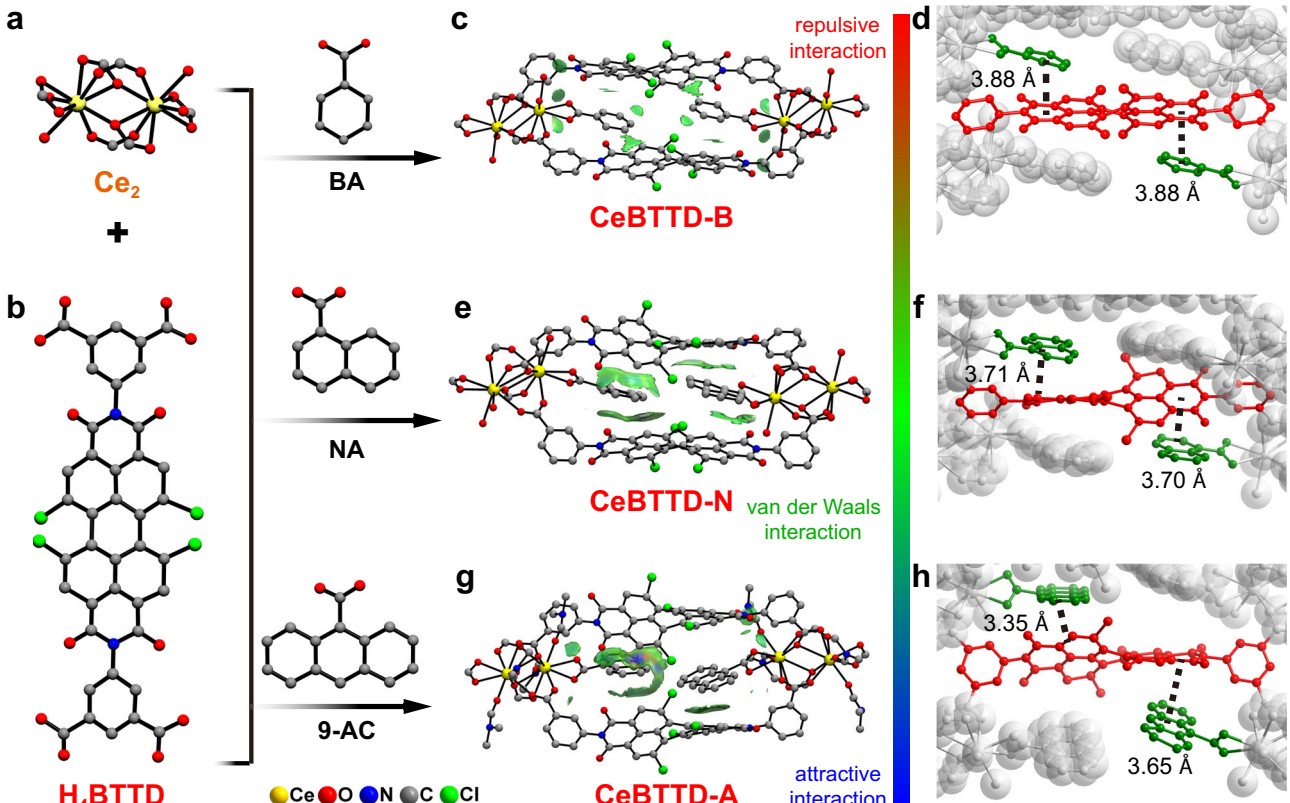

**Fig. 2 | Summary of the structure of CeBTTD compounds. a** The coordination environments of Ce$_2$ node. **b** The structure of H$_4$BTTD. Spatial arrangement and the independent gradient model based on Hirshfeld partition (IGMH) of **c** CeBTTD-B, **e** CeBTTD-N, **g** CeBTTD-A, respectively. Sign $(\lambda_2)\rho$ colored IGMH($\delta_g^{inter}$) = 0.004 a.u. isosurfaces. The coloring method is as follows: the green area corresponds to the van der Waals interaction ($\lambda_2 \approx 0$); the blue area corresponds to $\lambda_2 < 0$, and the attraction effect is stronger; the red area corresponds to the stronger repulsion ($\lambda_2 > 0$). The distance of π-π interaction in **d** CeBTTD-B, **f** CeBTTD-N, **h** CeBTTD-A, respectively. All hydrogen atoms are omitted for clarity. Ce, yellow; O, red; N, blue; C, gray; Cl, green.

higher than that of photocatalysts reported so far. More importantly, all the photocatalytic oxidation reactions can be effectively realized by CeBTTD-A under sunlight and ambient air, which shows the wide application prospect of CeBTTD-A.

## Results

### Preparation of CeBTTD-B/N/A molecular junction photocatalysts

The tetracarboxydiimide ligand (H$_4$BTTD) was synthesized by acylation reaction (see Methods). Then CeBTTD-B/N/A molecular junctions were further synthesized using a solvothermal method with Ce$_6$ cluster [Ce$_6$O$_4$(OH)$_4$(NH$_3$CH$_2$COO)$_8$(NO$_3$)$_4$(H$_2$O)$_6$Cl$_8$·8H$_2$O] precursor, H$_4$BTTD and three secondary ligands (BA, NA, and 9-AC) (Supplementary Fig. 2). Single-crystal X-ray diffraction analysis reveals that CeBTTD-B crystallizes in the monoclinic space group of $C2/c$, and the asymmetric unit contains one Ce$^{3+}$ ion, 1/2 BTTD$^{4-}$ ligand, one chelate coordinated BA, two coordinated water molecules (Supplementary Fig. 3). The Ce$_6$ cluster precursor was decomposed and recombined into a Ce$_2$ cluster [Ce$_2$(COO)$_6$(H$_2$O)$_4$] (Fig. 2a). The topological structure of CeBTTD-B was analyzed by TOPOS software. The Ce$_2$ cluster and BTTD$^{4-}$ (Fig. 2b) were regarded as 4-connected codes, respectively, which reveals the topology of CeBTTD-B simplified to *pts* topology, and the Schläfli symbol is {4$^2$·8$^4$}. Further analysis through three-dimensional structural (Supplementary Fig. 4) demonstrates that there is a weak π–π interaction between BA$^-$ and BTTD$^{4-}$ in CeBTTD-B with a 3.88 Å distance (Fig. 2c, d). CeBTTD-N has a similar structure to CeBTTD-B, except that NA replaces BA (Fig. 2e, Supplementary Figs. 5 and 6). Analogously, a stronger π–π interaction compared to CeBTTD-B between NA$^-$ and BTTD$^{4-}$ in 3.71 Å exists (Fig. 2f).

Two benzene rings of BTTD$^{4-}$ ligand in CeBTTD-N are coplanar, while those in CeBTTD-B are almost perpendicular (Supplementary Fig. 7a, b). The different skew angle of BTTD$^{4-}$ results in the CeBTTD-N framework is assigned into the default *lvt* topology with the symbol of Schläfli {4$^2$·8$^4$}. CeBTTD-A crystallizes in the tetragonal *Pnma* space group and shows an analogous coordination configuration (Supplementary Figs. 8 and 9). Due to the similar dihedral angle (Supplementary Fig. 7c) of the two benzene rings of BTTD$^{4-}$, CeBTTD-A reveals the same *lvt* topology as CeBTTD-N. The strongest conjugacy of 9-AC$^-$ results in the distance between 9-AC and BTTD$^{4-}$ as short as 3.35 Å (Fig. 2g, h), which can be favorable for the transfer of photogenerated charges between adjacent ligands.

The powder X-ray diffraction (PXRD) patterns of as-synthesized CeBTTD-B, CeBTTD-N, and CeBTTD-A are in accordance with the simulation curves (Supplementary Figs. 10–12), which confirmed the high purity of the obtained samples. The Fourier transform infrared spectroscopy (FTIR) spectra show that the three compounds contain similar components and functional groups (Supplementary Figs. 13). Thermogravimetric analysis (TGA) reveals that CeBTTD-B, CeBTTD-A, and CeBTTD-N have similar weight loss before 200 °C, which can be attributed to the departure of water and DMF solvent molecules. The skeletons can be stabilized up to 400 °C, which proves that these materials have good thermal stability (Supplementary Fig. 14). X-ray photoelectron spectroscopy (XPS) data shows the existence of elements (Ce, C, N, O, and Cl) in CeBTTD-B, CeBTTD-N, and CeBTTD-A (Supplementary Fig. 15a). High-resolution Ce 3$d$ spectra show two main peaks at 885.4 and 903.8 eV corresponding to Ce$^{3+}$. The same position of Ce 3$d$ peaks in CeBTTD-B, CeBTTD-N, and CeBTTD-A indicates that Ce ions have the same valence state (Supplementary Fig. 15b–d).

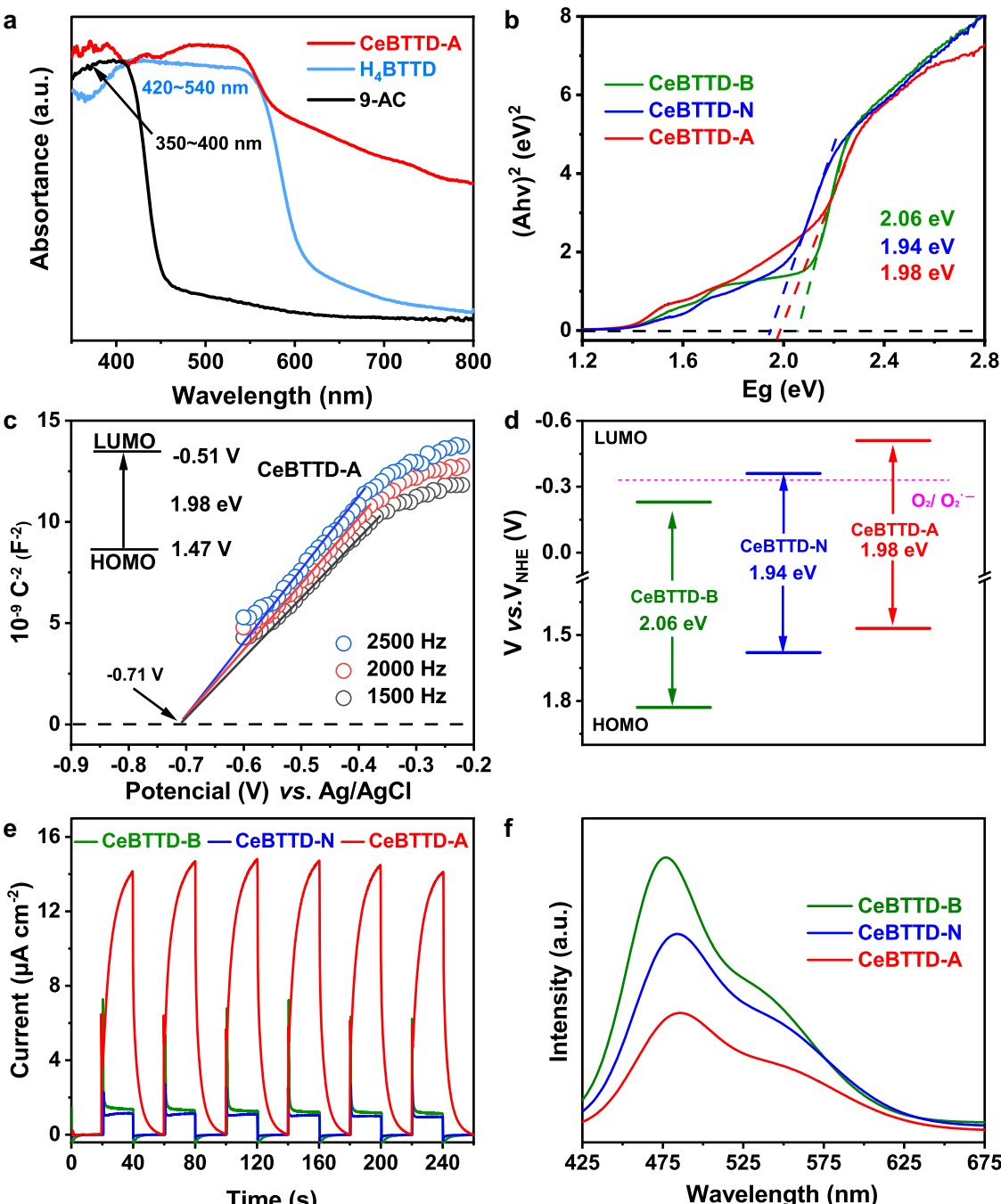

**Fig. 3 | Photophysical characterization of CeBTTD catalysts. a** UV–visible diffuse reflection spectra for CeBTTD-A and its motifs (9-AC and H₄BTTD). **b** Tauc plot of CeBTTD-(B/N/A) determined by the Kubelka–Munk formula from the original UV–visible diffuse reflection spectra. **c** Mott–Schottky plots of CeBTTD-A. **d** The energy band structures diagram for CeBTTD-(B/N/A). **e** Transient photocurrent curves of CeBTTD-(B/N/A) that were measured in 0.5 M Na₂SO₄ aqueous solution. **f** Photoluminescence (PL) emission spectra of CeBTTD-(B/N/A) under excitation at 400 nm.

The light absorption of three compounds and corresponding ligands was studied using UV–visible diffuse reflectance spectroscopy (Fig. 3a). It can be found that the light absorption of H₄BTTD is mainly in the range of 420–450 nm, and that of 9-AC is mainly in the range of 350–400 nm. Opportunely staggered light absorption ranges allow the two ligands to absorb different wavelengths of light and thus be excited without interfering and competing with each other. However, due to the weak light absorption capacity of BA and NA, the light absorption of CeBTTD-B and CeBTTD-N (Supplementary Figs. 16 and 17) in the ultraviolet region is weaker than CeBTTD-A. The band gaps of CeBTTD-A, CeBTTD-N, and CeBTTD-B were further calculated to be 1.98 eV, 1.94 eV, and 2.06 eV, respectively, which indicates their semiconductor-like characteristics (Fig. 3b). To determine the lowest unoccupied molecular orbital (LUMO) energy level positions, the Mott-Schottky plots test was performed at frequencies of 1500, 2000, and 2500 Hz (Fig. 3c, Supplementary Figs. 18 and 19). The results showed that the LUMO positions were determined to be −0.51 V (CeBTTD-A), −0.36 V (CeBTTD-N), and −0.23 V (CeBTTD-B) (vs. NHE, pH = 7), which shows that CeBTTD-A and CeBTTD-N can thermodynamically complete the O₂ activation process (O₂/O₂·⁻ −0.33 V, vs. NHE). Subsequently, based on the values of LUMO and band gaps, the highest orbital position (HOMO) of these three compounds was

calculated to be 1.47 V (CeBTTD-A), 1.58 V (CeBTTD-N), and 1.83 V (CeBTTD-B) (vs. NHE, pH = 7), respectively. The positive potential indicates that the photogenerated holes of the three compounds have strong oxidation capacity (Fig. 3d). The charge separation efficiency of the three photocatalysts was evaluated by the transient photocurrent response under periodic light illumination (Fig. 3e). The transient photocurrent response demonstrates that CeBTTD-A shows nearly seven times higher than CeBTTD-N or CeBTTD-B without attenuation, reflecting CeBTTD-A has the strongest separation and transfer efficiency of photogenerated charges. Furthermore, time-resolved fluorescence decay spectra were tested to evaluate the charge carrier dynamics. As depicted in Fig. 3f and Supplementary Fig. 20, CeBTTD-A has the lowest fluorescence intensity, revealing the best efficiency of photogenerated electron-hole separation among these compounds[27,28].

## Photocatalytic aerobic oxidation of C(sp³)−H bonds

Benefiting from the molecular junction structure, good light absorption, and suitable band structure, CeBTTD compounds are expected to be used as photocatalysts for photocatalytic aerobic oxidation reactions. Firstly, we focused on the oxidation of toluene to judge the photocatalytic performance of the CeBTTD catalysts. As preliminary attempts, 5 mg CeBTTD (−A/B/N) crystals were used as heterogeneous photocatalysts in a simple operating system containing 3 mL acetonitrile ($CH_3CN$) and 0.1 mmol toluene. The reaction was performed under an $O_2$ atmosphere, room temperature, and irradiation with a xenon lamp as the light source (300–1100 nm). The species and selectivity of the products were determined by gas chromatography–mass (GC–MS) and gas chromatography (GC) spectrometry (Supplementary Fig. 21). After a 24-h reaction, benzoic acid was detected as the main product. CeBTTD-A achieved an excellent benzoic acid yield of 98%, while CeBTTD-B and CeBTTD-N could not catalyze toluene under the same reaction conditions. Such a stark contrast reveals that 9-AC plays a crucial role during the photocatalytic process. To verify this idea, a series of contrast experiments were carried out (Supplementary Table 8). When the system was in the dark or lacked CeBTTD-A, toluene could not be converted, indicating that the reaction is a light-driven catalytic process. No product could be detected when $N_2$ replaces $O_2$ in the system, supporting the idea that oxygen is the oxidation agent. Interestingly, the physical mixture of $H_4BTTD$ and 9-AC also realized photooxidation of toluene with a yield of 70%, which was commensurate with the yield catalyzed by the physical mixture of $CeCl_3$, $H_4BTTD$ and 9-AC. These results indicate that Ce ions are not involved in the photocatalytic process. Since 9-AC is homogeneous in the reaction system, it can form molecular junction structures similar to that in CeBTTD-A through π−π interaction with $H_4BTTD$, thus achieving the migration of photogenic charges and showing photocatalytic activity. The higher performance obtained by heterogeneous CeBTTD-A further illustrates that the periodic distributed molecular junctions in CeBTTD-A are the key factor in promoting the photocatalytic aerobic oxidation of C(sp³)−H bonds. The substrate scope of toluene was investigated to demonstrate its general applicability (Supplementary Table 9). Various derivatives of toluene as substrates were tested under the same condition (Fig. 4a). When the substrate contained electron-withdrawing $p$-$OCH_3$, the acid yield could reach 99%. When the substrates contained electron-donating $p$-F, $p$-Cl, $p$-Br, and $p$-CN, the acid yields were determined to be 98%, 95%, 76%, and 62%, respectively. The substrates equipped with steric hindrance groups, such as $o$-bromotoluene were also tested; the yield of the corresponding acid was 55%. In addition, when the substrate is $p$-xylene, CeBTTD-A can obtain a 73% yield of terephthalic acid, which is an important and demanding chemical raw material. Notably, when the reaction was performed in ambient air, the performance did not decline significantly, demonstrating that CeBTTD-A possesses the ability to rapidly activate oxygen. Furthermore, the PXRD and FTIR

characterizations confirmed CeBTTD-A structural integrity before and after photocatalytic tests (Supplementary Figs. 22 and 23).

In order to verify the versatility of CeBTTD-A, another aerobic oxidation of the C(sp³)−H bond reaction, ethylbenzene oxidation, was further tested. Surprisingly, CeBTTD-A demonstrated excellent efficiency (>85%) and selectivity (>99%) in converting ethylbenzene derivatives into corresponding aryl ketones under either $O_2$ or ambient air atmospheres (Fig. 4b, Supplementary Fig. 24 and Supplementary Table 10). Besides the benzylic C−H bonds, we further tested the allylic C−H bond oxidation reaction using cyclohexene as the substrate. The result revealed that cyclohexene could be efficiently converted into 2-cyclohexen-1-one with a high conversion rate (>74%) and selectivity (>97%) under both $O_2$ and ambient air atmospheres, demonstrating the good universality of the CeBTTD-A photocatalyst (Supplementary Fig. 25). The durable, heterogeneous CeBTTD-A can be recycled at least six times without apparent loss of activity (Fig. 4c). Because of the high selectivity and efficient production of acetophenone, the catalytic durability of CeBTTD-A was also evaluated by the gram-scale reaction. A solvent-free 10-g-scale reaction of ethylbenzene oxidation under ambient air was carried out (Supplementary Fig. 26). After 10 days of light irradiation, 10.01 g ethylbenzene was completely transformed. 10.34 g isolated acetophenone with 91.4% yield was obtained after filtering the photocatalyst (Fig. 4d). Such excellent performance surpasses all catalysts reported up to now (Fig. 4e and Supplementary Table 11). The above satisfactory results prove that CeBTTD-A has a bright prospect in practical industrial applications.

## Discussion

In order to explore the catalytic mechanism, we first identified the active species in the reaction process. A series of quenching experiments were performed by adding 0.3 mmol quenching agent to the original test solution of toluene/ethylbenzene oxidation (Fig. 4f and Supplementary Fig. 27). The results are similar and as follows. When a hole quencher (KI) or electron quencher ($AgNO_3$) was added, only a trace of the products could be detected after 4-h light irradiation, suggesting that photogenerated electrons and hole radicals are engaged in the reaction. When a $^1O_2$ radical quencher (L-Histidine, L-HIS) was introduced, there was no significant change in reaction performance. While the $O_2^{·−}$ radical quenching agent ($p$-benzoquinone, BQ) is added, the catalytic efficiency was significantly weakened, stating $O_2$ is reduced to $O_2^{·−}$ and involved in the oxidation reaction. To verify the generation of the above-mentioned ROS, in-situ electron paramagnetic resonance (EPR) spectroscopy of CeBTTD-A was performed. As shown in Fig. 5a, under the oxygen atmosphere, there is no apparent signal under dark conditions. After 30 min of illumination, the superoxide radical ($O_2^{·−}$) signals appeared in the system with the presence of DMPO (5,5- Dimethyl-1-pyrroline N-oxide), indicating that CeBTTD-A could rapidly activate oxygen molecules to superoxide radicals under light irradiation. $^1O_2$ was also detected in the presence of TEMP (2,2,6,6-Tetramethyl-4-piperidone hydrochloride).

Transient absorption spectra (TAS) tests were performed to determine the process of photoexcitation and excited state species generation of each component in CeBTTD-A (Fig. 5b, c), which can help to analyze the transition mode of photogenerated charges. The positive excited state absorption (ESA) bands of 9-AC and negative ground-state bleaching (GSB) peak of CeBTTD-A at 550 nm can explain that the photogenerated electrons of 9-AC can quickly migrate to other motifs in CeBTTD-A (Supplementary Fig. 28). The negative stimulated emission in $H_4BTTD$ at 650 nm is weakened in CeBTTD-A, indicating that the photoelectrons from 9-AC are transferred to $BTTD^{4−}$. The unchanged electron density of Ce ions determined by in situ XPS under light irradiation showed that Ce ions do not receive photoelectrons (Supplementary Fig. 29). The positive ESA bands at 1100 nm show that the lifetime of excited state 9-AC is much longer than $BTTD^{4−}$ and CeBTTD-A, which is in line with the analysis of kinetic processes

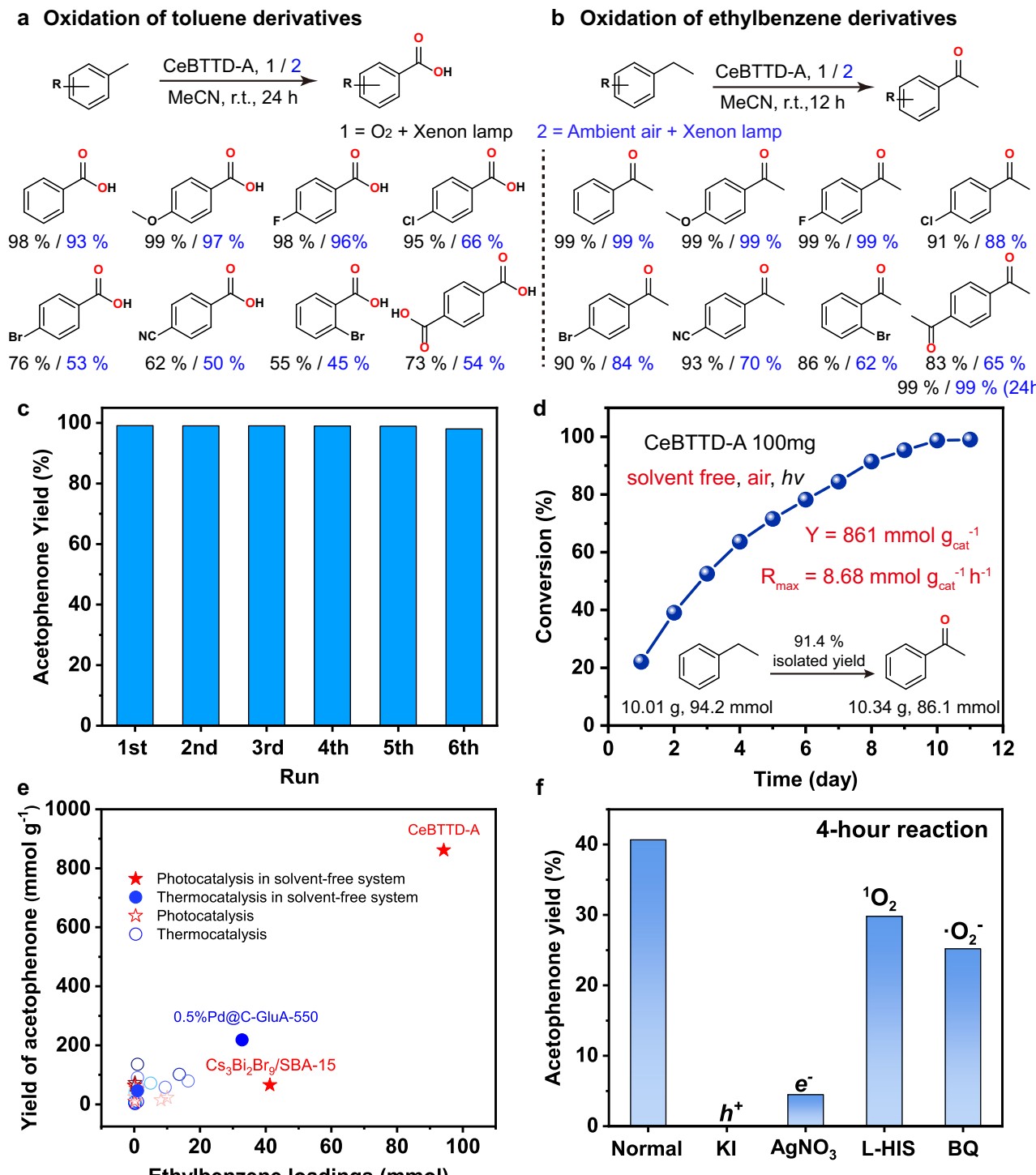

**Fig. 4 | The photocatalytic performance of CeBTTD-A for aerobic oxidation of C(sp³)-H bonds. a** CeBTTD-A catalyzed the aerobic oxidation of toluene derivatives. **b** CeBTTD-A catalyzed the aerobic oxidation of ethylbenzene derivatives. Reaction conditions: 0.1 mmol substrates, 5 mg CeBTTD-A, 3 mL CH₃CN as the reagent, 300–1100 nm xenon light source. **c** Cycle experiment of aerobic oxidation of ethylbenzene along with CeBTTD-A as photocatalyst. **d** Kinetic profile for solvent-free ten-gram reaction for oxidation of ethylbenzene. **e** Comparison of catalytic performance between CeBTTD-A and other advanced catalysts in photo- or thermal-driven systems. **f** Control experiments for photocatalytic ethylbenzene oxidation by CeBTTD-A under normal conditions or with different scavengers.

probed at 1100 nm[29,30]. It also proves that the migration direction of the photoelectron is from 9-AC to BTTD⁴⁻. Density functional theory (DFT) theoretical calculations were performed to better understand the photogenerated charge separation and migration process of CeBTTD-A under illumination. As shown in Supplementary Fig. 30, the main vertical excitations of CeBTTD-A under illumination occur at 397,

412, 415, 484, and 547 nm. The attribution of the electron transition indicates that the excitation near 400 nm is mainly contributed by the transition of 9-AC itself (Supplementary Table 12), which is consistent with the absorption spectrum of 9-AC (Fig. 3a). Among them, the major contributions at 397, 412, and 484 nm are the electrons in HOMO-1 of 9-AC excited to other orbits (LUMO + 4, LUMO + 2 and LUMO + 1

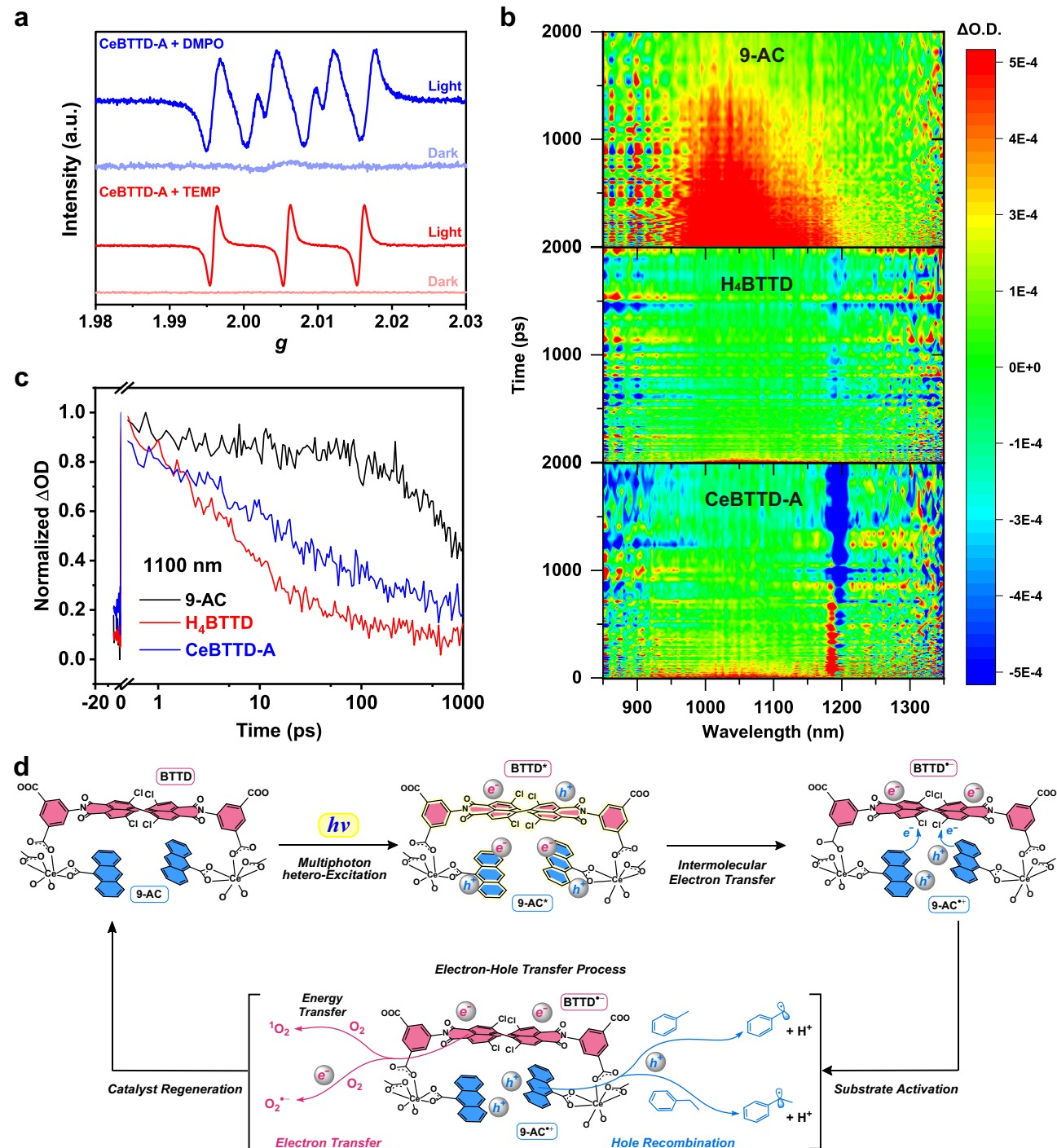

**Fig. 5 | Study on photoexcitation process and photocatalysis mechanism. a** EPR signals of the reaction solution under the dark and light irradiation in the presence of DMPO and TEMP as the spin-trapping reagents. **b** TA spectra of 9-AC, H₄BTTD, and CeBTTD-A were probed within the region of 850–1350 nm. Excitation wavelength: 9-AC (340 nm), H₄BTTD (400 nm), and CeBTTD-A (400 nm). **c** The kinetic traces of 9-AC, H₄BTTD, and CeBTTD-A were probed at 1100 nm. **d** Proposed mechanism for aerobic oxidation of C(sp³)−H bonds by CeBTTD-A.

respectively, Supplementary Fig. 31). It means that the hole located in HOMO-1 with higher energy than HOMO can be the major part involving the toluene C−H bond oxidation. Combined with the XAS characterization results, a reasonable photogenerated charge separation and transition process was proposed (Fig. 5d). Firstly, some electrons located in different orbits of 9-AC and BTTD motifs are excited and generate corresponding BTTD* and 9-AC* excited species under illumination. Subsequently, the excited photoelectrons of 9-AC* are transferred to BTTD*, forming 9-AC⁺ and BTTD⁻. The photogenerated

holes located in the inner orbits of 9-AC complete the oxidation of the C-H bonds, and the electrons on BTTD complete the O₂-to-O₂⁻ reduction and finally achieve the oxidation of C(sp³)−H bonds.

In addition to the aerobic oxidation of C(sp³)−H bonds, we broaden our testing to other photocatalytic aerobic reactions, such as benzyl alcohol oxidation, anisole oxidation, and benzylamine oxidative coupling reactions to assess the universal applicability of CeBTTD-A. As shown in Fig. 6a−c, all these aerobic reactions demonstrated excellent photocatalytic performance and good substrate tolerance.

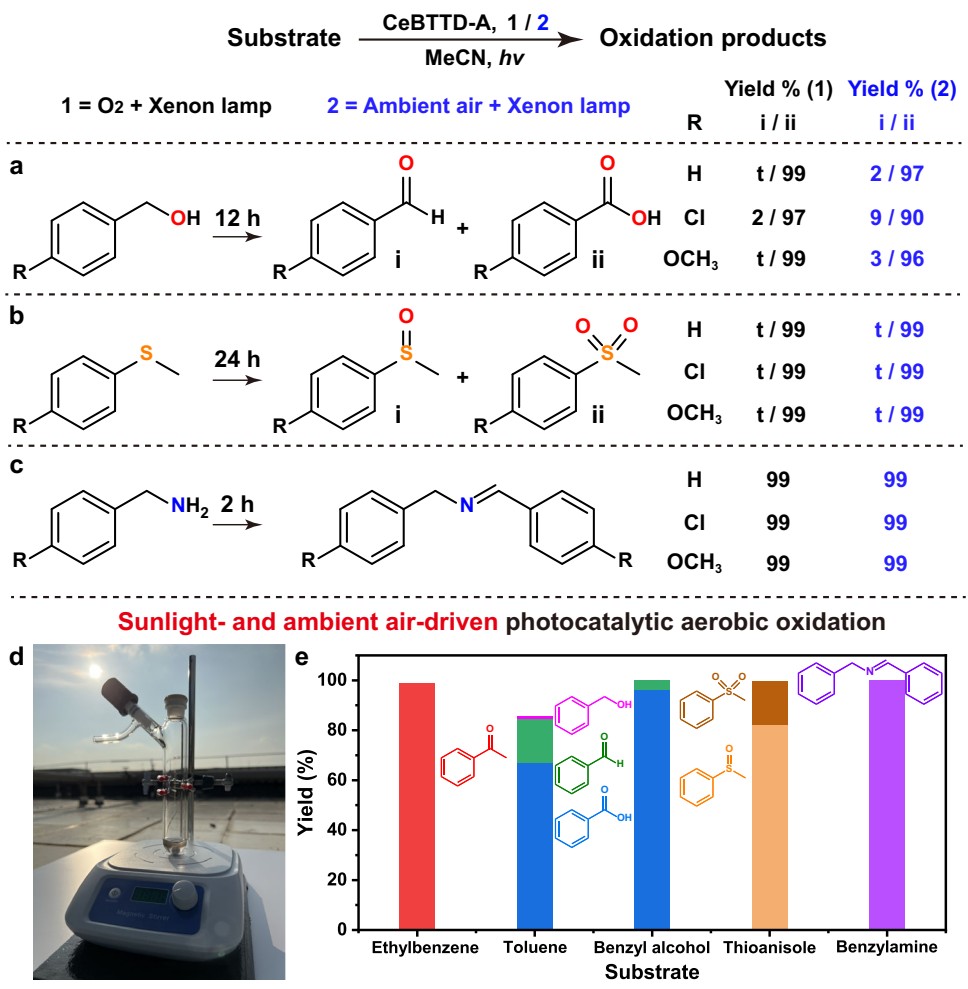

**Fig. 6 | Photocatalytic performance in other photocatalytic aerobic reactions and sunlight- and ambient air-driven aerobic oxidation reactions.** CeBTTD-A catalyzed **a** the oxidation of benzyl alcohol derivatives, **b** the oxidation of thioanisole derivatives, **c** benzylamine derivatives coupling reaction, respectively. Reaction condition: 0.1 mmol substrates, 5 mg CeBTTD-A, 3 mL CH₃CN as the reagent, 300–1100 nm xenon light source, 25.0 °C, t = trace. **d** The facility of photocatalytic experiments in the outdoor environment and **e** products yields for sunlight- and ambient air-driven aerobic oxidation reactions. Reaction condition: 0.1 mmol substrates, 5 mg CeBTTD-A, 3 mL CH₃CN as the reagent, sunlight, ambient air, 8:00–16:00, Guangzhou.

These promising results encourage us to perform the five photocatalytic oxidation reactions mentioned in this paper in outdoor sunlight and ambient air. When the reactions were executed under real sunlight on October 10–12, 2022 (Guangzhou, China), the oxidation reactions could still be completed after 8-h illumination (Fig. 6d). Different from the results of indoor laboratory reactions, the product of the oxidation of thioanisole was mainly methyl phenyl sulfoxide, which is probably caused by the shorter duration of illumination (Fig. 6e) than indoor laboratory condition. This highly efficient photocatalytic system driven entirely by sunlight and air without any additional cost further demonstrated the advantages and commercial potential of CeBTTD-A.

In summary, we designed and developed a hetero-motif molecular junction as a model photocatalyst applied to photocatalytic aerobic oxidation of C(sp³)−H bonds. Benefited from the strong π-π interaction between the motifs in the molecular junction structure, the constructed photocatalyst CeBTTD-A can undergo rapid intermolecular migration of photogenerated charges and substrate (O₂ and C(sp³)-H bonds) activation under light irradiation, so as to achieve high reactivity for oxidation of toluene and ethylbenzene derivatives. A solvent-free ten-gram-scale reaction of ethylbenzene oxidation under ambient air could be fully accomplished, which reveals better performance than other photo- or thermal-driven catalytic systems reported up to now. Apart from C(sp³)-H bond oxidation reactions, CeBTTD-A

can also be applied to three other kinds of general aerobic reactions and also exhibit excellent reaction activities. More importantly, all these aerobic reactions can be driven entirely by sunlight and air without additional energy or cost input, which reveals the commercialized application prospect. We anticipate that our operating principles of hetero-motif molecular junction photocatalyst will provide inspiration to develop next-generation photocatalysts for widespread applications in constructing complex molecular blocks and break the bottleneck of sunlight- and ambient air-driven photocatalysis of more challenging reactions in an atom-economic and environment-friendly and sustainable way.

## Methods

### Synthesis

All starting materials, reagents, and solvents used in experiments were commercially available, high-grade purity materials and used without further purification.

### Preparation of H₄BTTD

N,N-Bis(3,5-Dicarboxyl)-1,6,7,12-Tetrachloroperylene-3,4,9,10-Tetracarboxylic Diimide (H₄BTTD, BTTD in the manuscript represents the deprotonated ligand): In a 50 mL single necked flask, a mixture of 3,5-dibromoaniline (3.0 g, 12.0 mmol), 1,6,7,12-Tetrachloroperylene Tetracarboxylic Acid Dianhydride (1.0 g, 2.0 mmol) and propanoic acid

(25.0 mL) was heated in 160 °C with a reflux condenser and a magnetic stirring bar for 48 h. After the reaction, the organic solvent was removed, and the precipitate was collected under reduced pressure at room temperature. Wash the precipitate with plenty of water and a small amount of ethanol to remove residual propionic acid. The crude product was dried in a vacuum oven at 60 °C for 9 h, and red powder $H_4BTTD$ was obtained as pure product (yield 80% based on dianhydride). $^1$H-NMR (400 MHz, D$_6$-DMSO): δ = 8.30 (d, $^3J$ = 1.6 Hz, 4H, *CH*), 8.60 (t, $^3J$ = 1.6 Hz, 2H, *CH*), 8.64 (s, 4H, *CH*) ppm.

## Preparation of Ce$_6$

$Ce_6O_4(OH)_4(NH_3CH_2COO)_8(NO_3)_4(H_2O)_6Cl_8·8H_2O$: Ce$_6$ is prepared based on previous reports with slight modifications[31]. Ammonium cerium nitrate (30.0 g, 54.7 mmol) and glycine (9.0 g, 120.0 mmol) were dissolved in 30.0 mL ultrapure water to obtain an orange solution. Then, the solution was added to 270.0 mL of a saturated sodium chloride solution. Then, the obtained yellow solution was placed at 25 °C for a week, and the solids were filtered and collected. The solid was washed with a small amount of ice water solution several times to remove the excessive glycine and sodium chloride. The crude product was dried in a vacuum oven in 60 °C for 5 h, and the light-yellow solid Ce$_6$ was obtained (70% based on ammonium cerium nitrate).

## Preparation of CeBTTD-B

$Ce_2BTTD(C_6H_5COO)_2(H_2O)_4$: 30 mg Ce$_6$, 5 mg H$_4$BTTD, 20 mg Benzoic acid (BA for short), and 3 mL DMF were charged in a 10 mL pressure-resistant glass tube and heat at 150 °C for 72 h. After cooling down to room temperature, orange single crystals of CeBTTD-B were collected (85% based on H$_4$BTTD).

## Preparation of CeBTTD-N

$Ce_2BTTD(C_{10}H_9COO)_2(H_2O)_4$: 30 mg Ce$_6$, 5 mg H$_4$BTTD, 20 mg 1-naphthoic acid (NA for short) and 3 mL DMF were charged in a 10 mL pressure-resistant glass tube and heat at 150 °C for 72 h. After cooling down to room temperature, brown single crystals of CeBTTD-N were collected (70% based on H$_4$BTTD).

## Preparation of CeBTTD-A

$Ce_2BTTD(9\text{-}AC)_2(DMF)_2(H_2O)_2$: 30 mg Ce$_6$, 5 mg H$_4$BTTD, 20 mg 9-anthroic acid (9-AC for short), and 3 mL DMF were charged in a 10 mL pressure-resistant glass tube and heat at 150 °C for 72 h. After cooling down to room temperature, dark brown single crystals of CeBTTD-A were collected (80% based on H$_4$BTTD).

## Single-crystal X-ray diffraction

Single-crystal X-ray intensity data collection for CeBTTD crystals was carried out at 150 K with a Bruker D8 Venture diffractometer equipped with a PHOTON III detector using the monochromatized wavelength (Cu Kα) = 1.54178 Å. The crystal structures were solved and refined by full-matrix least-squares methods against $F^2$ using the SHELXL-2014 program package[32,33] and Olex-2 software[34]. All non-hydrogen atoms were refined with anisotropic displacement parameters, and hydrogen positions were fixed at calculated positions and refined isotropically. The topological analysis was performed with the TOPOS program[35]. The crystallographic data and structure refinement for CeBTTD crystals are summarized in Supplementary Table 1. The selected bond lengths and bond angles of CeBTTD-B, CeBTTD-N, and CeBTTD-A are listed in Supplementary Tables 2–7, respectively.

Due to the small size of CeBTTD-B and CeBTTD-N crystals, the diffraction intensity was weak, resulting in low resolution and low bond precision on C–C bonds. The B-level alerts of "D–H bond without acceptor" can be explained that the acceptor might be the free solvent molecules squeezed. Although there are some problems in these X-ray analyses, we think that these results can strongly support the

determination of the structure of CeBTTD coordination polymers and the discussion about the bond lengths and angles.

## Analytical techniques

Thermogravimetric analysis (TGA) of the sample was performed on a Perkin-Elmer TG-7 analyzer heated from room temperature to 800 °C in flowing N$_2$/O$_2$ with a heating rate of 10 °C/min. Fourier transform infrared spectroscopy (FTIR) was recorded in the range of 4000–400 cm$^{-1}$ on a Mattson Alpha-Centauri spectrometer using the technique of pressed KBr pellets. Powder X-ray diffraction (PXRD) measurements were recorded ranging from 5 to 50° at room temperature on a D/max 2500 VL/PC diffractometer (Japan) equipped with graphite mono-chromatized Cu Kα radiation (λ = 1.54060 Å)[24]. The UV–visible diffuse reflection spectra were acquired on a Shimadzu UV-2550 spectrophotometer in the wavelength range of 300–800 nm. X-ray photoelectron spectroscopy (XPS) was measured on an Escalab 250Xi. In situ electron paramagnetic resonance (EPR) spectra were recorded on a Bruker A300 spectrometer. Electrochemical measurements (photocurrent and the Mott–Schottky spots) were carried out using an electrochemical workstation CHI 660E. The products of the aerobic oxidation reaction were detected and analyzed by gas chromatography-mass spectrometer (GC–MS, 8890-5977B, Agilent), gas chromatography (GC, 8890, Agilent), and nuclear magnetic resonance ($^1$H NMR, Varian VNMRS).

The femtosecond transient absorption setup used for this study is based on a regenerative amplified Ti:sapphire laser system from Coherent (800 nm, 35 fs, 6 mJ pulse-1, and 1 kHz repetition rate), nonlinear frequency mixing techniques and the Femto-TA100 spectrometer (Time-Tech Spectra). Briefly, the 800 nm output pulse from the regenerative amplifier was split into two parts with a 50% beam splitter. The transmitter part was used to pump a TOPAS Optical Parametric Amplifier (OPA), which generates a wavelength-tunable laser pulse from 250 nm to 2.5 μm as a pump beam. The reflected 800 nm beam was split again into two parts. One part with less than 10% was attenuated with a neutral density filter and focused into a 2 mm thick sapphire window to generate a white light continuum (WLC) from 420 nm to 800 nm used for the probe beam. The probe beam was focused with an Al parabolic reflector onto the sample. After the sample, the probe beam was collimated and then focused into a fiber-coupled spectrometer with CMOS sensors and detected at a frequency of 1 kHz[36]. The delay between the pump and probe pulses was controlled by a motorized delay stage. The pump pulses were chopped by a synchronized chopper at 500 Hz, and the absorbance change was calculated with two adjacent probe pulses (pump-blocked and pump-unblocked). Nanosecond TA measurement was performed with the EOS spectrometer (Ultrafast Systems LLC). The pump beam is generated in the same way as the femtosecond TA experiment described above. A different white light continuum (380–1700 nm, 0.5 ns pulse width, 20 kHz repetition rate) was used, which was generated by focusing a Nd:YAG laser into a photonic crystal fiber. The delay time between the pump and probe beam was controlled by a digital delay generator (CNT-90, Pendulum Instruments)[37].

## Photocatalysis activity measurement

Before the catalytic tests, the single crystal of CeBTTD-(B/N/A) was washed with DMF and ethanol in turn and then heated at 60 °C for 4 h in a vacuum oven.

Three milliliter CH$_3$CN (GR) and 5 mg catalysts are placed in a 50 mL quartz tube. A flow rate of 0.1 L/min for 5 min under the gas (high pure N$_2$/O$_2$ > 99.995%) to ensure that the air in the reaction system was completely exchanged (when the atmosphere is ambient air, this step is omitted). Then, 0.1 mmol substrate was quickly added to the tube under the gas flow. A 300 W Xenon lamp (light intensity: 200 mW cm$^{-2}$, λ = 300–1100 nm) was used as the light source. The

reaction temperature was controlled at 298 K by using the cooling water circulation. After the reaction, the solution was collected, centrifuged, and filtered through a 0.22 μm syringe filter to remove catalyst particles. In the photocatalytic cycle tests, the photocatalysts were centrifuged and washed with $CH_3CN$ for the next tests.

## Electrochemical measurement

Electrochemical tests were made in 0.5 M $Na_2SO_4$ solution as the electrolyte through the standard three-electrode system: indium–tin oxide (ITO) glass modified with catalyst samples, carbon rod, and saturated Ag/AgCl used as the working electrode, counter electrode, and the reference electrode, respectively. The working electrodes for photocurrent and Mott-Schottky spots were prepared as follows: 2 mg photocatalyst were dispersed in a mixed solution of 950 μL ethanol and 50 μL Nafion D-520 dispersion solutions to generate a homogeneous slurry. Then, 200 μL the slurry was transferred and coated on ITO glass plates (1 cm × 2 cm) and dried at room temperature.

## Femtosecond transient absorption spectroscopy (TAS) measurement

The fs-TAS measurements were conducted on a pump-probe system (Helios, Ultrafast Systems LLC) with a maximum time delay of ~8 ns controlled by a motorized optical delay line. An 800-nm ultrashort laser pulse was generated by an amplified Ti:sapphire laser system (Coherent Legend, 800 nm, 85 fs, 7 mJ/pulse, and 1 kHz repetition rate) and was split into two beams. The main beam was frequency-doubled through a beta-barium borate (BBO) crystal to 400 nm and acted as the pump pulse, and neutral-density filters were utilized to control the power at ~160 μW. The other beam was transformed into a white-light continuum (420–800 nm) via a sapphire to become the probe pulse, which probed the states of the specimen after photoexcitation by the pump pulse[38].

## Calculation methods

All of the geometry optimizations were performed at the level of density functional theory (DFT) using Gaussian 09 program[39] at the (U) B3LYP[40]/[6-31 G(d)[41]/SDD[42](Ce)] level. The absorption spectra and excited state species were obtained by using time-dependent DFT (TDDFT) and unrestricted open-shell DFT at the same level. The orbital overlap integral was calculated by the Amsterdam Density Functional (ADF 2023.101)[43] program package. The zero-order regular approximation (ZORA) was adopted to account for the scalar relativistic effects[44]. The basis sets for describing all electrons of each atom are triple-$\zeta$ plus polarization Slater-type orbitals[45].

## Data availability

Crystallographic data reported in this paper is provided in the Supplementary Information and archived at the Cambridge Crystallographic Data Center under reference number CCDC 2267826–2267828 (CeBTTD-B, CeBTTD-N, and CeBTTD-A). These data can be obtained free of charge from The Cambridge Crystallographic Data Center via "www.ccdc.cam.ac.uk/data_request/cif". All the data supporting the findings of this study are available within the article and its Supplementary Information. Source data are provided in this paper.

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

## Acknowledgements

This work was financially supported by the National Key R&D Program of China (2023YFA1507204 to Y.-Q.L.), the National Natural Science Foundation of China (22225109 to Y.-Q.L., No. 22201082 to L.Z., 22301086 to X.-X.L., 22301084 to R.-H.L.), the GuangDong Basic and Applied Basic Research Foundation (No. 2021A1515110429 to L.Z.), and project funded by China Postdoctoral Science Foundation (No. 2022M721216 to L.Z., 2023T160235 to L.Z.).

## Author contributions

Y.-Q.L. conceived the idea and designed the experiments. X.-X.L. and L.Z. synthesized the materials and carried out the photocatalytic experiments. R.-H.L. carried out the DFT simulation for the photocatalytic reaction mechanism. L.Z., X.-X.L., S.W., J.L., X.-X.H., L.-Z.D. and S.-L.L. analyzed the data. L.Z., X.-X.L., R.-H.L. and Y.-Q.L. interpreted the results and wrote the paper.

## Competing interests

The authors declare no competing interests.
