## [Peer Review File · Nature Communications]

Photocatalytic aerobic oxidation of C(sp³)-H bondsREVIEWER COMMENTS

Reviewer #1 (Remarks to the Author):

Currently, it is still a major challenge to realize nature-driven photocatalysis, in this manuscript submitted by Lan et al., a series of hetero-motif molecular junction (HMMJ) photocatalysts are constructed and can be used to the direct oxidation of C(sp³)-H bonds and various aerobic oxidation reactions. It is very interesting that these photocatalytic reactions can also be conducted under nature-drive conditions with satisfied yield and selectivity. Furthermore, the catalytic mechanism is also analyzed in detail. This work shows a potential way to design photocatalysts, which can break the bottleneck of nature-driven photocatalysis of more challenging reactions. In my view, it can be considered for publication in Nature Communication. Some advices are given as follows:

1. The different photocatalytic performances of CeBTDD-B, CeBTDD-N and CeBTDD-A are attributed to the different π - π interactions by the authors, as shown in Figure 2. However, due to the perylene diimide ligand adopted in this work is an electron-deficient molecule (acceptor) while 9-anthraic acid is the best electron-rich molecule (donor) compared to benzoic acid and naphthalene acid. Could the photoinduced electron transfer gifted by the donor-acceptor pairs in CeBTDD-A contribute to its best photocatalytic activity?

2. As description in the manuscript, Ce ions are not involved in the photocatalytic process. How about replacing Ce with other lanthanide ions?

3. The assignments of Ce3d peaks in supplementary Figure 15b is a little rough. Please see refs 10.1039/C4CC10346G and 10.1002/cssc.202102368.

4. Some minor mistakes should be corrected such as:

In line 238, "electron-donating" should be "electron-withdrawing".

Line 241, "Bromotoluene" should be "bromotoluene".

Reviewer #3 (Remarks to the Author):

In this manuscript, the authors described a photocatalytic hetero-motif molecular junction mediated aerobic oxidation of C(sp³)-H bonds to yield the corresponding carboxylic acids and ketone products. The heterogeneous catalysts in this study contains cerium cation, a perylene diimide ligand and a 9-anthraic acid conjugated ligand. The catalytic system is highly efficient under xenon lamp and O₂ atmosphere, as well as air and ambient sunlight. The heterogeneity of the catalyst enables rapid removal and recycle the catalysts, which grants high practicality of the catalytic system. I recommend acceptance of this manuscript, if the authors can address the following questions.

1. The HOMO of CeBTDD-A is 1.47 V vs. SCE. However, the oxidation potential of toluene has been reported to be as high as 2.4 V vs. SCE (Synlett 2016, 27, 714–723). How is the photogenerated hole of CeBTDD-A capable of oxidation of toluene, given the ~1 V endothermic free energy change of the proposed oxidation step.

2. Besides the benzylic C–H bonds, have the authors tried allylic C–H bonds, such as cyclohexene, or cinnamic alcohol?

Reviewer #4 (Remarks to the Author):

Lan et al. report “Nature-driven Aerobic Oxidation of C(sp³)-H Bonds”. In this manuscript, a series of CeBTDD HMMJs were synthesized, characterized and catalyzed aerobic oxidation of C(sp³)-H bonds. In particular, solvent-free large-scale ethylbenzene oxidation and nature-driven aerobic oxidation reactions were realized, which is very important and meaningful. In view of its novelty and sufficient data, this work can be considered for publication in Nature Communications after minor revisions. The following points need to be deemed to improve the manuscript.

1. More detailed experimental procedures should be provided. From the current manuscript, it remains unclear about the crystal pretreatment and photocatalytic cycle tests (Figure 4c).
2. The topology diagrams should be depicted to clearly understand the structures of CeBTDD HMMJs.
3. In the description of the photocatalytic performance, it is better to provide the standard curves detected by GC, which is the basis of all photocatalytic data.
4. In Supplementary Figure 14, the TGA curves of CeBTDD-B and CeBTDD-N showed that the temperature dropped slightly (~ 450 °C). It should be explained or retested.
5. The absorption curve represented by the different colored lines is not clear in Supplementary Figure 27.
6. When revising the manuscript, authors are recommended to pay particular attention to some minor mistakes. For example, the aerobic oxidation reaction properties with terephthalic acid as the substrate are missing from Supplementary Table 9; p-F is an electron-acceptor group rather than an electron-donor group.

Reviewer #1 (Remarks to the Author):

Currently, it is still a major challenge to realize nature-driven photocatalysis, in this manuscript submitted by Lan et al., a series of hetero-motif molecular junction (HMMJ) photocatalysts are constructed and can be used to the direct oxidation of C(sp³)-H bonds and various aerobic oxidation reactions. It is very interesting that these photocatalytic reactions can also be conducted under nature-drive conditions with satisfied yield and selectivity. Furthermore, the catalytic mechanism is also analyzed in detail. This work shows a potential way to design photocatalysts, which can break the bottleneck of nature-driven photocatalysis of more challenging reactions. In my view, it can be considered for publication in Nature Communication. Some advices are given as follows:

1. The different photocatalytic performances of CeBTDD-B, CeBTDD-N and CeBTDD-A are attributed to the different π - π interactions by the authors, as shown in Figure 2. However, due to the perylene diimide ligand adopted in this work is an electron-deficient molecule (acceptor) while 9-anthracic acid is the best electron-rich molecule (donor) compared to benzoic acid and naphthalene acid. Could the photoinduced electron transfer gifted by the donor-acceptor pairs in CeBTDD-A contribute to its best photocatalytic activity?

Response: Thank you very much for your kind suggestion. In fact, we initially considered the situation you mentioned. However, X-ray transient absorption characterization (Figure 5) and DFT calculation about frontier molecular orbital results show that the photogenerated electron excitation and transition modes in CeBTDD-A are quite different from those in classic D-A pairs. Generally, in a D-A structure, only one of the motifs is photoexcited (D-A* type or D*-A type). However, in CeBTDD-A, BTDD and 9-AC are both excited under light irradiation and generate 9-AC* and BTDD*. Then the charge migration and recombination occur from 9-AC* to BTDD*, thereby retaining the photogenerated electrons on BTDD^{•-} and the holes of 9-AC^{•+}. This charge transition mode is similar to semiconductor heterojunctions, which can greatly facilitate the separation of photogenerated charges. In the comparison, the

photocatalytic performance of CeBTDD-N decreased significantly because the 1-naphthalene acid (NA) cannot be excited by light (Supplementary Figure 17).

2. As description in the manuscript, Ce ions are not involved in the photocatalytic process. How about replacing Ce with other lanthanide ions?

Response: Thank you very much for your important suggestion. According to your suggestion, we synthesized two other two isomorphous photocatalysts EuBTDD-A and NdBTDD-A (Figures R1 and R2), and applied them to aerobic toluene oxidation and ethylbenzene oxidation reactions. The test results showed that their photocatalytic performance is almost indistinguishable from that of CeBTDD-A (Tables R1 and R2), which further verifies that Ce ions are not involved in the photocatalytic process.

Figure R1. The experimental PXRD patterns of EuBTDD-A (green), NdBTDD-A (blue), CeBTDD-A (red) and simulated PXRD pattern of CeBTDD-A (black), respectively.

Figure R2. **a** XPS survey spectrum and **b** High-resolution peaks of Eu 3d spectrum of EuBTDD-A, **c** XPS survey spectrum and **d** High-resolution peaks of Nd 3d spectrum of NdBTTD-A, respectively.

Table R1. The photocatalytic performance of oxidation of toluene with different photocatalysts.

Photocatalyst	Conversion (%)	Benzaldehyde Selectivity (%)	Benzoic acid Selectivity (%)
CeBTDD-A	99	Trace	99
EuBTDD-A	98	Trace	99
NdBTTD-A	95	Trace	99

Reaction condition: 5 mg photocatalyst, 0.1 mmol substrate, 24 hours, 3 mL CH₃CN as the reagent, 300 W xenon light irradiation (300-1100 nm), 1 atm O₂.

Table R2. The photocatalytic performance of oxidation of ethylbenzene with different photocatalysts.

Photocatalyst	Conversion (%)
CeBTTD-A	99
EuBTTD-A	99
NdBTTD-A	99

Normal condition: 5 mg photocatalyst, 0.1 mmol substrate, 12 hours, 3 mL CH₃CN as the reagent, 300 W xenon light irradiation (300-1100 nm), 1 atm O₂.

3.The assignments of Ce 3d peaks in supplementary Figure 15b is a little rough. Please see refs 10.1039/C4CC10346G and 10.1002/cssc.202102368.

Response: Thanks very much for your helpful suggestion. We have reprocessed Ce 3d XPS spectra. Please see the revised Supplementary Materials (Supplementary Figure 15)

Supplementary Figure 15. a XPS survey spectra and High-resolution peaks of Ce 3d spectrum of b CeBTDD-B, c CeBTDD-N and d CeBTDD-A, respectively. There is no satellite peak located around 916.0 eV, which suggest that Ce ions in CeBTDD HMMJs are all in +3 oxidation state.¹

1. Bêche, E., *et al.* Ce 3d XPS investigation of cerium oxides and mixed cerium oxide ($Ce_xTi_yO_z$). *Surf. Interface Anal.*, **40**, 264-267 (2008).

4. Some minor mistakes should be corrected such as:

In line 238, “electron-donating” should be “electron-withdrawing”.

Line 241, “Bromotoluene” should be “bromotoluene”.

Response: Thanks very much for your kind suggestion. We have corrected the statements. Please see the revised Manuscript.

When the substrate contained electron-withdrawing *p*-OCH₃, the acid yield could reach 99%. When the substrates contained electron-donating *p*-F, *p*-Cl, *p*-Br and *p*-CN, the acid yields were determined to be 98%, 95%, 76% and 62%, respectively.

The substrates equipped with steric hindrance groups, such as *o*-bromotoluene were also tested, the yield of the corresponding acid was 55%.

Reviewer #2

This manuscript by Lan and co-workers reports the construction of heteromotif molecular junction photocatalysts that combined two bifunctional photosensitive-activation motifs with strong interaction. For the prepared HMMJ photocatalyst CeBTDD-A, the reduction of O₂ and the oxidation of the C(sp³)-H bond can be efficiently accomplished by two photoactive units (H₄BTDD and 9-AC). Importantly, the strong interaction and short distance between H₄BTDD and 9-AC promote the rapid migration of photogenerated charge and mass transfer of reaction intermediates. Therefore, CeBTDD-A showed high efficiency in the oxidation of toluene to benzoic acid and ethylbenzene to acetophenone. This comprehensive work has revealed bicomponent excitation pathway to promote the rapid charge transfer and separation to form the electron-hole pairs and then to activate oxygen and C(sp³)-H bond under light illumination. The manuscript is well organized with solid data to support their conclusions. Overall, this paper is an interesting work, but a couple of issues should be addressed before it acceptable for publication. Details are given below.

Q1: Abstract, page 1, line 10. The authors mention that “synthesized a series of HMMJ photocatalysts that combined two bifunctional photosensitive-activation motifs with strong interaction. In the obtained HMMJ photocatalysts, the reduction of O₂ and the oxidation of C(sp³)-H bonds can be effectively accomplished by two motifs Thus, efficient aerobic oxidation of C(sp³)-H bonds can be realized.” In fact, for these HMMJ photocatalysts, only CeBTDD-A has high catalytic efficiency. Therefore, the authors should improve these expressions in the abstract to make them consistent with the main text.

Response: Thanks very much for your kind suggestion. We have modified the relevant statements to make them consistent with the main text. Please see the revised Manuscript file, Abstract part.

Especially in the obtained HMMJ photocatalysts, CeBTDD-A, the reduction of O₂ and the oxidation of C(sp³)-H bonds can be effectively accomplished by two motifs. The

strong interaction and short distance between them promote the rapid migration of photogenerated charge and mass transfer of reaction intermediates ($\text{O}_2^{\bullet-}$ and $\text{Ar-CH}^{\bullet+}$ -R, R = H or CH_3). Thus, efficient aerobic oxidation of $\text{C}(\text{sp}^3)$ -H bonds can be realized.

Q2: Introduction, page 4, line 13. The authors mention that “a series of HMMJ photocatalysts (CeBTDD-B CeBTDD-N and CeBTDD-A) were assembled with a perylene diimide ligand (H4BTDD) (Supplementary Figure 1) and three conjugated ligands (benzoic acid (BA), 1-naphthalene acid (NA) and 9-anthroic acid (9-AC)) to achieve efficient nature-driven aerobic oxidation of $\text{C}(\text{sp}^3)$ -H bonds reactions.” There is also the same problem, that is, only CeBTDD-A has high catalytic efficiency. According to the information you have given, CeBTDD-B and CeBTDD-N cannot drive this reaction. So, the authors should change the corresponding expression.

Response: Thanks very much for your helpful suggestion. We have changed the corresponding expression for accuracy. Please see the revised Manuscript file, page 4.

Based on the above consideration, a series of HMMJ photocatalysts (CeBTDD-B CeBTDD-N and CeBTDD-A) were assembled with a perylene diimide ligand (H4BTDD) (Supplementary Figure 1) and three conjugated ligands (benzoic acid (BA), 1-naphthalene acid (NA) and 9-anthroic acid (9-AC)). Due to the restriction of the coordination orientation of the Ce_2 secondary building unit (SBU), the BTDD^{4+} and the secondary ligands exhibit short spatial distances ($3.3 \sim 3.8 \text{ \AA}$) and strong π - π interaction in the structure of CeBTDD photocatalysts. When three HMMJ photocatalysts were used for the aerobic oxidation of $\text{C}(\text{sp}^3)$ -H bond reactions, CeBTDD-A showed high efficiency in the nature-driven oxidation of toluene to benzoic acid and ethylbenzene to acetophenone with yields in excess of 98%.

Q3: Results, page 7, line 14. “the Mott-Schottky plots test was performed at frequencies of 1500-2500 Hz.” This sentence should be described as “the Mott-Schottky plots test was performed at frequencies of 1500, 2000, and 2500 Hz.”

Response: Thanks very much for your kind suggestion. We have revised the corresponding description. Please see the revised Manuscript file, page 7.

To determine the lowest unoccupied molecular orbital (LUMO) energy level positions, the Mott-Schottky plots test was performed at frequencies of 1500, 2000 and 2500 Hz (Figure 3c, Supplementary Figures 18 and 19).

Q4: The authors should give the full name before abbreviations appear, such as PL in the captions of figure 3f.

Response: Thanks very much for your helpful suggestion. We have added the full name of PL in the captions of Figure 3f and Supplementary Figure 20. Please see the revised manuscript and Supplementary Materials.

f Photoluminescence (PL) emission spectra of CeBTDD HMMJs under excitation at 400 nm.

Supplementary Figure 20. Photoluminescence (PL) emission spectra of CeBTDD HMMJs and ligands under excitation at 400 nm.

Q5: Results, page 9, line 5. In addition to wavelength information, the type of light source should also be given.

Response: Thank you very much for your important suggestion. We have added the type of light source and revised the statement. Please see the revised Manuscript,

The reaction was performed under O₂ atmosphere, room temperature and irradiation with a xenon lamp as the light source (300-1100 nm).

Q6: About Figure 4e. In addition to showing the catalytic efficiency of CeBTDD-A, the authors also listed many reported heterogeneous materials capable of aerobic oxidation of ethylbenzene. The annotations given in the figure 4e indicate that Cs₃Bi₂Br₉/SBA-

15 belongs to thermocatalysis system. However, in Supplementary Table 11, Cs₃Bi₂Br₉/SBA-15 belongs to photocatalytic system. The authors should confirm the accuracy of the quoted information.

Response: Thank you very much for your helpful suggestion. After our verification, Cs₃Bi₂Br₉/SBA-15 indeed belongs photocatalytic system. We have corrected the Figure 4e. Please see the revised Manuscript, page 10.

Figure 4. The photocatalytic performance of CeBTDD-A for aerobic oxidation of C(sp³)-H bonds. **a** CeBTDD-A catalyzed the aerobic oxidation of toluene derivatives. **b** CeBTDD-A catalyzed the aerobic oxidation of ethylbenzene derivatives. Reaction

conditions: 0.1 mmol substrates, 5 mg CeBTDD-A, 3 mL CH₃CN as the reagent, 300-1100 nm xenon light source. **c** cycle experiment of aerobic oxidation of ethylbenzene along with CeBTDD-A as photocatalyst. **d** kinetic profile for solvent-free ten-gram reaction for oxidation of ethylbenzene. **e** comparison of catalytic performance between CeBTDD-A and other advanced catalysts in photo- or thermal-driven systems. **f** control experiments for photocatalytic ethylbenzene oxidation by CeBTDD-A under normal conditions or with different scavengers.

Q7: The authors believe that the strong π - π interaction between two active units plays a crucial role in the catalytic reaction. Therefore, it is suggested that the authors should add the catalytic efficiency of single ligand in the controlled experiment to further verify the conclusions.

Response: Thank you very much for your important suggestion. We have tested the photocatalytic performance of single ligand and added the data into Supplementary Table 8 and Supplementary Table 10.

Supplementary Table 8. The photocatalytic performance of toluene oxidation with different photocatalysts.

Entry	Photocatalyst	Reaction Condition	Conversion (%)
1	CeBTDD-A	normal	99
2	CeBTDD-N	normal	trace
3	CeBTDD-B	normal	trace
4	CeBTDD-A	no substrate	trace
5	CeBTDD-A	N ₂ replace O ₂	trace
6	no catalyst	normal	trace
7	CeBTDD-A	dark	trace
8	9-AC (homogeneous photocatalyst)	normal	48
9	H ₄ BTDD	normal	trace
10	H ₄ BTDD : 9-AC (2 mg : 1 mg)	normal	70
11	H ₄ BTDD : 9-AC : CeCl ₃ (2 mg : 1 mg : 2 mg)	normal	72

Normal condition: 5 mg photocatalyst, 0.1 mmol substrate, 24 hours, 3 mL CH₃CN as

the reagent, 300 W xenon light irradiation (300-1100 nm), 1 atm O₂.

Supplementary Table 10. The photocatalytic performance of ethylbenzene oxidation with different photocatalysts.

Entry	Photocatalyst	Reaction Condition	Conversion (%)
1	CeBTTD-A	normal	99
2	CeBTTD-N	normal	1
3	CeBTTD-B	normal	trace
4	CeBTTD-A	no substrate	trace
5	CeBTTD-A	N ₂ replace O ₂	trace
6	no catalyst	normal	trace
7	CeBTTD-A	dark	trace
8	9-AC (homogeneous photocatalyst)	normal	35
9	H ₄ BTTD	normal	trace
10	H ₄ BTTD : 9-AC (2 mg : 1 mg)	normal	65
11	H ₄ BTTD : 9-AC : CeCl ₃ (2 mg : 1 mg : 2 mg)	normal	63

Normal condition: 5 mg photocatalyst, 0.1 mmol substrate, 12 hours, 3 mL CH₃CN as the reagent, 300 W xenon light irradiation (300-1100 nm), 1 atm O₂.

Q8: The English writing could be improved.

Response: Thank you very much for your kind suggestion. We have carefully modified some sentences in the manuscript to improve its readability.

Reviewer #3 (Remarks to the Author):

In this manuscript, the authors described a photocatalytic hetero-motif molecular junction mediated aerobic oxidation of C(sp³)-H bonds to yield the corresponding carboxylic acids and ketone products. The heterogeneous catalysts in this study contains cerium cation, a perylene diimide ligand and a 9-anthroic acid conjugated ligand. The catalytic system is highly efficient under xenon lamp and O₂ atmosphere, as well as air and ambient sunlight. The heterogeneity of the catalyst enables rapid removal and recycle the catalysts, which grants high practicality of the catalytic system. I recommend acceptance of this manuscript, if the authors can address the following questions.

1. The HOMO of CeBTDD-A is 1.47 V vs. SCE. However, the oxidation potential of toluene has been reported to be as high as 2.4 V vs. SCE (Synlett 2016, 27, 714–723). How is the photogenerated hole of CeBTDD-A capable of oxidation of toluene, given the ~1 V endothermic free energy change of the proposed oxidation step.

Response: Thank you very much for your very helpful suggestion. Based on your suggestion, we researched the relevant references and found that the currently determined HOMO of CeBTDD-A is indeed unable to complete the single-electron oxidation of toluene. However, a series of test results show that the photocatalyst CeBTDD-A we constructed exhibits excellent photocatalytic performance for toluene oxidation under both xenon lamp and ambient sunlight. Therefore, a reasonable explanation is that C-H bond oxidation of toluene is completed in an orbit with higher energy than HOMO. To support the explanation, DFT calculations were performed to better understand the photogenerated charge separation and migration process of CeBTDD-A under illumination. As shown in Supplementary Figure 30, the main vertical excitations of CeBTDD-A under illumination occur at 397, 412, 415, 484 and 547 nm. The attribution of the electron transition indicates that the excitation near 400 nm is mainly contributed by the transition of 9-AC itself, which is consistent with the absorption spectrum of 9-AC (Figure 3a). Among them, the major contributions at 397,

412 and 484 nm are the electrons in HOMO-1 of 9-AC excited to other orbits (LUMO+4, LUMO+2 and LUMO+1 respectively). It means that the hole located in HOMO-1 with higher energy can be the major part involving the toluene C-H bond oxidation. Combined with the XAS characterization results, a more reasonable photogenerated charge separation and transition process was proposed. Under illumination, some electrons located in different orbits of 9-AC and BTDD motifs are excited. Subsequently, the excited photoelectrons of 9-AC* are transferred to BTDD*, forming 9-AC^{•+} and BTDD^{•-}. The photogenerated holes located in the inner orbits of 9-AC complete the oxidation of the C-H bonds, and the electrons on BTDD complete the O₂-to-O₂^{•-} reduction, and finally achieve the oxidation of toluene to benzoic acid. Due to the involvement of inner orbital energy levels, the change of Gibbs free energy values (ΔG) in original manuscript was unreliable. Therefore, we removed these values in the revised manuscript, Figure 5d.

Density functional theory (DFT) theoretical calculations were performed to better understand the photogenerated charge separation and migration process of CeBTDD-A under illumination. As shown in Supplementary Figure 30, the main vertical excitations of CeBTDD-A under illumination occur at 397, 412, 415, 484 and 547 nm. The attribution of the electron transition indicates that the excitation near 400 nm is mainly contributed by the transition of 9-AC itself (Supplementary Table 12), which is consistent with the absorption spectrum of 9-AC (Figure 3a). Among them, the major contributions at 397, 412 and 484 nm are the electrons in HOMO-1 of 9-AC excited to other orbits (LUMO+4, LUMO+2 and LUMO+1 respectively, Supplementary Figure 31). It means that the hole located in HOMO-1 with higher energy than HOMO can be the major part involving the toluene C-H bond oxidation. Combined with the XAS characterization results, a reasonable photogenerated charge separation and transition process was proposed (Figure 5d). Firstly, some electrons located in different orbits of 9-AC and BTDD motifs are excited and generate corresponding BTDD* and 9-AC* excited species under illumination. Subsequently, the excited photoelectrons of 9-AC* are transferred to BTDD*, forming 9-AC^{•+} and BTDD^{•-}. The photogenerated holes located in the inner orbits of 9-AC complete the oxidation of the C-H bonds, and the electrons on BTDD complete the O₂-to-O₂^{•-} reduction, and finally achieve the oxidation of C(sp³)-H bonds.

Figure 5. Study on photoexcitation process and photocatalysis mechanism. a EPR signals of the reaction solution under the dark and light irradiation in the presence of DMPO and TEMP as the spin-trapping reagents. **b** TA spectra of 9-AC, H₄BTDD and CeBTDD-A probed within the region of 850-1350 nm. Excitation wavelength: 9-AC (340 nm), H₄BTDD (400 nm) and CeBTDD-A (400 nm). **c** the kinetic traces of 9-AC, H₄BTDD and CeBTDD-A probed at 1100 nm. **d** proposed mechanism for aerobic oxidation of C(sp³)-H bonds by CeBTDD-A.

Supplementary Figure 30. Calculated absorption spectrum of CeBTTD-A.

Supplementary Table 12. Absorption peaks λ_{abs} [in eV (nm)], the oscillator strengths (f), major contributions and assignments (H and L represent HOMO and LUMO, respectively).

λ_{abs}	f	Major Contributions	Major Assignments
547	0.428	H-3 \rightarrow L (96%),	BTTD \rightarrow BTTD
484	0.031	H-1 \rightarrow L+1 (49%)	9-AC \rightarrow BTTD
415	0.062	H \rightarrow L+2 (76%)	9-AC \rightarrow 9-AC
412	0.074	H-1 \rightarrow L+2 (78%)	9-AC \rightarrow 9-AC
397	0.042	H-1 \rightarrow L+4 (86%)	9-AC \rightarrow 9-AC

Supplementary Figure 31. The calculated major frontier molecular orbital distributions involved in the excitation process of Supplementary Table 11.

2. Besides the benzylic C-H bonds, have the authors tried allylic C-H bonds, such as cyclohexene, or cinnamic alcohol?

Response: Thank you very much for your important suggestion. We have tested the allylic C-H bond oxidation reaction with CeBTDD-A as the photocatalyst. When the substrate was cyclohexene or cinnamic alcohol, both oxidation processes could occur with high conversion rates (83% for cyclohexene and 94% for cinnamic alcohol respectively) under the O₂ atmosphere (Figure R3). The major product of cyclohexene oxidation was 2-cyclohexen-1-one with 97% selectivity. When the substrate was cinnamic alcohol, in addition to cinnamic aldehyde, the product also contained nearly the same amount of benzaldehyde. Additionally, the oxidation reactions of cyclohexene and cinnamic alcohol can be also efficiently accomplished in an air atmosphere (Figure R4), which further demonstrates the high activity and universality of the hetero-motif molecular junction photocatalyst we constructed.

Figure R3. Photocatalytic performance for aerobic oxidation of cyclohexene and cinnamic alcohol. Reaction condition: 0.1 mmol substrates, 5 mg CeBTDD-A as the photocatalyst, 1 atm O₂, 24 hours, 3 mL CH₃CN as the reagent, 300-1100 nm xenon light source.

Figure R4. Photocatalytic performance for aerobic oxidation of cyclohexene and cinnamic alcohol. Reaction condition: 0.1 mmol substrates, 5 mg CeBTDD-A as the photocatalyst, air as reaction atmosphere, 24 hours, 3 mL CH₃CN as the reagent, 300-1100 nm xenon light source.

We have added the relevant data into the revised Manuscript and Supplementary Materials.

Besides the benzylic C-H bonds, we further tested the allylic C-H bond oxidation reaction using cyclohexene as the substrate. The result revealed that cyclohexene could be efficiently converted into 2-cyclohexen-1-one with high conversion rate (> 74%) and selectivity (> 97%) under both O₂ and ambient air atmospheres, demonstrating the outstanding universality of the CeBTDD-A photocatalyst (Supplementary Figure 25).

Supplementary Figure 25. Photocatalytic performance for aerobic oxidation of cyclohexene. Reaction condition: 0.1 mmol substrates, 5 mg CeBTDD-A as the photocatalyst, air or O₂ as reaction atmosphere, 24 hours, 3 mL CH₃CN as the reagent, 300-1100 nm xenon light source.

Reviewer #4 (Remarks to the Author):

Lan et al. report “Nature-driven Aerobic Oxidation of C(sp³)-H Bonds”. In this manuscript, a series of CeBTDD HMMJs were synthesized, characterized and catalyzed aerobic oxidation of C(sp³)-H bonds. In particular, solvent-free large-scale ethylbenzene oxidation and nature-driven aerobic oxidation reactions were realized, which is very important and meaningful. In view of its novelty and sufficient data, this work can be considered for publication in Nature Communications after minor revisions. The following points need to be deemed to improve the manuscript.

1. More detailed experimental procedures should be provided. From the current manuscript, it remains unclear about the crystal pretreatment and photocatalytic cycle tests (Figure 4c).

Response: Thank you very much for your kind suggestion. We have described the details of crystal pretreatment and photocatalytic cycle tests in our revised Manuscript. Please see the revised Manuscript file, Photocatalysis activity measurement part.

Before the catalytic tests, the single crystal of CeBTDD HMMJs were washed with DMF and ethanol in turn and then heated at 60 °C for 4 hours in a vacuum oven.

In the photocatalytic cycle tests, the photocatalysts were centrifuged and washed with CH₃CN for the next tests.

2. The topology diagrams should be depicted to clearly understand the structures of CeBTDD HMMJs.

Response: Thank you very much for your insightful suggestion. We have added the topology diagrams of CeBTDD HMMJs in our revised Supplementary Materials (Supplementary Figures 4, 6 and 9).

Supplementary Figure 4. **a** The coordination environment of Ce₂ in CeBTDD-B. Each Ce₂ in CeBTDD-B was coordination with four BTDD⁴⁻, two BA⁻ and two coordinated water molecules. All hydrogen atoms are omitted for clarity. Ce, yellow, Cl, green, O, red, C, gray, N, blue. **b** The three-dimensional structural and **c** the corresponding topology diagrams of CeBTDD-B, which the yellow balls represent {Ce₂} and black balls represent BTDD⁴⁻, respectively.

Supplementary Figure 6. a The coordination environment of Ce₂ in CeBTDD-N. Each Ce₂ in CeBTDD-N was coordination with four BTDD⁴⁺, two NA⁻ and two coordinated water molecules. All hydrogen atoms are omitted for clarity. Ce, yellow, Cl, green, O, red, C, gray, N, blue. **b** The three-dimensional structural and **c** the corresponding topology diagrams of CeBTDD-N, which the yellow balls represent {Ce₂} and black balls represent BTDD⁴⁺, respectively.

Supplementary Figure 9. **a** The coordination environment of Ce₂ in CeBTDD-A. All hydrogen atoms are omitted for clarity. Ce, yellow, Cl, green, O, red, C, gray, N, blue. **b** The three-dimensional structural and **c** the corresponding topology diagrams of CeBTDD-A, which the yellow balls represent {Ce₂} and black balls represent BTDD⁴⁻, respectively.

3. In the description of the photocatalytic performance, it is better to provide the standard curves detected by GC, which is the basis of all photocatalytic data.

Response: Thanks very much for your good suggestion. According to your suggestion, we have provided the standard curves detected by GC in our revised Supplementary Materials (Supplementary Figures 21 and 24).

Supplementary Figure 21. Standard curves of **a** toluene, **b** benzyl alcohol, **c** benzaldehyde and **d** benzoic acid by GC analysis.

Supplementary Figure 24. Standard curves of **a** ethylbenzene and **b** acetophenone by GC analysis.

4. In Supplementary Figure 14, the TGA curves of CeBTDD-B and CeBTDD-N showed that the temperature dropped slightly (~ 450 °C). It should be explained or retested.

Response: Thank you very much for your insightful suggestion. We have retested the TGA curves of CeBTDD-B and CeBTDD-N. Please see the revised Supplementary Materials (Supplementary Figure 14).

Supplementary Figure 14. The TGA curves of CeBTDD-B (green), CeBTDD-N (blue) and CeBTDD-A (red), respectively.

5. The absorption curve represented by the different colored lines is not clear in Supplementary Figure 27.

Response: Thank you very much for your helpful suggestion. We have modified Supplementary Figure 30. Please see the Supplementary Materials.

Supplementary Figure 30. Calculated absorption spectrum of CeBTDD-A.

6. When revising the manuscript, authors are recommended to pay particular attention to some minor mistakes. For example, the aerobic oxidation reaction properties with terephthalic acid as the substrate are missing from Supplementary Table 9; *p*-F is an electron-acceptor group rather than an electron-donor group.

Response: Thank you very much for your kind suggestion. We are sorry for these mistakes. According to your suggestion, we modified the related statements in the revised Manuscript and Supplementary Materials.

When the substrate contained electron-withdrawing *p*-OCH₃, the acid yield could reach 99%. When the substrates contained electron-donating *p*-F, *p*-Cl, *p*-Br and *p*-CN, the acid yields were determined to be 98%, 95%, 76% and 62%, respectively.

Supplementary Table 9. Photocatalytic performance of oxidation of toluene derivatives with CeBTDD-A as the photocatalyst.

R	Atmosphere	Yield (%)		Conversion (%)
		R-ph-CHO	R-ph-COOH	
p -H	O ₂	trace	98	98
	Air	2	93	95
p -OCH ₃	O ₂	trace	99	99
	Air	trace	97	97
p -CN	O ₂	10	62	72
	Air	7	50	57
p -F	O ₂	trace	98	98
	Air	2	96	98
p -Cl	O ₂	3	95	98
	Air	19	66	85
p -Br	O ₂	13	76	89
	Air	22	53	75
o -Br	O ₂	22	55	77
	Air	10	45	55
COOH-ph-COOH				
p -CH ₃	O ₂	trace	trace	74
	Air	trace	trace	53

Condition: 5 mg photocatalyst, 0.1 mmol substrate, 24 hours, 3 mL CH₃CN as the reagent, 300 W xenon light irradiation (300-1100 nm).

REVIEWERS' COMMENTS

Reviewer #1 (Remarks to the Author):

Now, the manuscript has been improved, which I think can be accepted for publish.

Reviewer #2 (Remarks to the Author):

The authors clearly responded to the referee's comments. The paper has been revised well and now can be accepted in the present form.

Reviewer #3 (Remarks to the Author):

I am glad to find out that the authors have acknowledged my two questions and made appropriate changes to both the manuscript and the SI. Therefore, I would like to recommend publication of this work on Nature Communications.

Reviewer #4 (Remarks to the Author):

I am satisfied with the revision to the manuscript. I agree on the publication of the manuscript in current state in Nature Communications.

One typo in Figure 1c, 'interanction' should be corrected as 'interaction'.

Reviewer #1:

Now, the manuscript has been improved, which I think can be accepted for publish.

Reviewer #2:

The authors clearly responded to the referee's comments. The paper has been revised well and now can be accepted in the present form.

Reviewer #3:

I am glad to find out that the authors have acknowledged my two questions and made appropriate changes to both the manuscript and the SI. Therefore, I would like to recommend publication of this work on Nature Communications.

Reviewer #4:

I am satisfied with the revision to the manuscript. I agree on the publication of the manuscript in current state in Nature Communications.

One typo in Figure 1c, 'interanction' should be corrected as 'interaction'.

Response: Thank you very much for your kind suggestion. We have corrected the spelling of the word. Please see the revised Manuscript.

Aerobic Oxidation of C(sp³)-H Bonds

a. Traditional industrial synthesis method

b. Desired photocatalytic synthesis method

c. Design ideas of photocatalyst (this work)

Figure 1. Catalysts for direct oxidation of C(sp³)-H bonds. **a** industrial aerobic oxidation processes using transition-metal-catalysts. **b** aerobic oxidation processes in a desired photocatalytic synthesis method. **c** design ideas of hetero-motif molecular junction photocatalyst for photocatalytic aerobic oxidation of C(sp³)-H bonds.